# Regenerating Damaged Myocardium: A Review of Stem-Cell Therapies for Heart Failure

**DOI:** 10.3390/cells10113125

**Published:** 2021-11-11

**Authors:** Dihan Fan, Hanrong Wu, Kaichao Pan, Huashan Peng, Rongxue Wu

**Affiliations:** 1Psychiatric Genetics Group, McGill University, Montreal, QC H4H 1R3, Canada; di.han.fan@umontreal.ca (D.F.); hanrong.wu@mail.mcgill.ca (H.W.); Huashan.peng@douglas.mcgill.ca (H.P.); 2Department of Psychiatry, McGill University, Montreal, QC H4H 1R3, Canada; 3Division of Cardiology, Department of Medicine, University of Chicago, Chicago, IL 60637, USA; kpan5@uchicago.edu

**Keywords:** iPSCs, ESC, differentiation, cardiovascular disease, myocardial repair

## Abstract

Cardiovascular disease (CVD) is one of the contributing factors to more than one-third of human mortality and the leading cause of death worldwide. The death of cardiac myocyte is a fundamental pathological process in cardiac pathologies caused by various heart diseases, including myocardial infarction. Thus, strategies for replacing fibrotic tissue in the infarcted region with functional myocardium have long been a goal of cardiovascular research. This review begins by briefly discussing a variety of somatic stem- and progenitor-cell populations that were frequently studied in early investigations of regenerative myocardial therapy and then focuses primarily on pluripotent stem cells (PSCs), especially induced-pluripotent stem cells (iPSCs), which have emerged as perhaps the most promising source of cardiomyocytes for both therapeutic applications and drug testing. We also describe attempts to generate cardiomyocytes directly from cardiac fibroblasts (i.e., transdifferentiation), which, if successful, may enable the pool of endogenous cardiac fibroblasts to be used as an in-situ source of cardiomyocytes for myocardial repair.

## 1. Introduction

The regenerative capacity of the human heart is extremely limited. Thus, myocardial tissue that is lost to acute or chronic ischemic injury cannot be replaced via endogenous mechanisms of repair, and since dysfunction within the scar damages cells at the scar perimeter, the infarcted region grows as the patient ages, which typically leads to heart failure. Unhealthy dietary choices, lack of exercise, and the general aging of the population have combined to increase the prevalence of heart failure in the US from 5.7 million to 6.2 million between 2013 and 2016, and heart failure contributed to 13.4% of US deaths in 2018 and a loss of $30.7 billion in 2012 [1]. Notably, the morbidity, mortality, and economic costs of heart failure are likely to be exacerbated by the current COVID-19 pandemic, because acute cardiac injury is commonly associated with SARS-CoV-2 infection [2].

Although current medical therapies can provide symptomatic relief and delay onset of the most severe complications of heart failure, sustained improvement in cardiac performance requires surgical interventions such as the installation of a left ventricular assist device or (eventually) heart transplantation. However, the long-term use of cardiac prosthetics is associated with a substantial risk of endocarditis, and the number of patients that would benefit from heart transplantation surgery far exceeds the availability of donated hearts. Thus, strategies for replacing fibrotic tissue in the myocardial scar with functional contractile tissue have long been a goal of cardiovascular research, and early (but occasionally controversial) evidence that stem/progenitor cells in tissues from the heart and other organs can promote myocardial recovery by (in part) differentiating into cardiac cells has spurred numerous potential therapeutic advancements, such as the administration of stem/progenitor cells directly to the heart, reprogramming somatic cells into stem-like cells (i.e., induced-pluripotent stem cells [iPSCs]), the in-vitro differentiation of stem/progenitor cells into cardiomyocytes before administration, and the fabrication of engineered heart tissues for placement over the myocardial scar. This review provides a brief overview of a variety of stem and progenitor cells that have been investigated for use in regenerative myocardial therapy (Table 1) [3,4,5,6,7,8,9] before focusing primarily on embryonic stem cells (ESCs) and iPSCs, which have emerged as perhaps the most promising sources of cardiomyocytes for both therapeutic applications and drug testing.

## 2. Somatic Stem and Progenitor Cells

### 2.1. Skeletal Myoblasts

Skeletal myoblasts are derived from satellite cells located beneath the basal lamina of muscle fibers and were the first stem-like cells used for clinical trials of cardiac regeneration [10]. Autologous skeletal myoblasts are easily obtained via muscle biopsy, and the cells can be rapidly expanded in vitro, are resistant to ischemia, and have low tumorigenic potential [4]. However, transplanted skeletal myoblasts have been associated with a high risk of ventricular arrhythmia in several clinical trials [11] likely because they fail to form the gap junctions needed to support electromechanical coupling between cardiomyocytes [3].

### 2.2. Bone Marrow (BM)-Derived Cells

In response to injury, cells are recruited from the BM to the damaged region, where they promote tissue repair and regeneration [12]. The mobilized BM cells include a variety of stem-cell populations, and the magnitude of BM stem-cell mobilization is positively correlated with improvements in cardiac function [13]. Thus, a number of cell-mobilizing agents, such as granulocyte colony-stimulating factor (G-CSF), have been used to increase the number of circulating stem cells in patients with myocardial infarction, but the results have been inconsistent, and meta-analyses found little evidence of benefit [14].

The direct administration of BM-derived cells to injured hearts was evaluated in a murine model of myocardial infarction as early as 2001 [3,15], and the improvements associated with transplanted BM cells in this and other animal studies led to clinical trials of BM mononuclear cells (MNCs) in patients. BM MNCs are heterogenous and include three main stem-cell types: hematopoietic stem cells (HSCs), endothelial progenitor cells (EPCs) and mesenchymal stem cells (MSCs) [5]. EPCs are a provasculogenic subpopulation of HSCs that express CD133 and other lineage markers, and the clinical translation of autologous CD133+ stem cells has progressed as far as phase 3. Patients who were undergoing coronary artery bypass graft surgery for treatment of chronic ischemic heart disease were administered CD133+ BM cells or placebo via direct intramyocardial injection during the surgical procedure. Six months later, measurements of left ventricular ejection fraction (LVEF) (cells: 31 ± 7%, placebo: 33 ± 8%, *p* = 0.3; inter-group difference: −2.1%, 95% CI −6.3 to 2.1) did not differ significantly between the two treatment groups, and although CD133+ cell administration was associated with greater improvements in myocardial perfusion (cells: 9%, placebo: 2%, *p* < 0.001) and scar mass (cells: 2 g reduction, placebo: unchanged, 95% CI −1.1 to 5) [16], these benefits dissipated over several years [17].

Unlike HSCs and EPCs, MSCs are immunosuppressive and do not induce an inflammatory response [3], which suggests that they may be suitable for allogeneic transplantation [18]; however, while the results from some randomized clinical trials suggest that MSC transplantation can improve left ventricular performance [19] others reported no benefit [20,21,22]. More recently, BM MSCs have been combined with MSCs that have been genetically engineered to continuously secrete hepatocyte growth factor (HGF-eMSCs) and suspended in 3-dimensional, printed patches [23]. This strategy promoted the viability and vasculogenic potential of the BM-MSCs, and when evaluated in infarcted rat hearts, measurements of LVEF were significantly greater, and cardiomyocytes were significantly more common, in animals treated with patches containing both BM-MSCs and HGF-eMSCs than with patches containing either individual MSC population or in untreated animals.

### 2.3. Adipose-Derived Stem Cells (ASCs)

Adipose-derived stem cells (ASCs) are multipotent, can be maintained in vitro for extended periods without inducing senescence, and express many of the same lineage markers present in MSCs [24,25,26]. Nevertheless, although ASCs are sometimes referred to as “adipose-derived MSCs,” they are different from the MSCs present in other tissues; for example, BM MSCs express the surface marker CD146, while ASCs do not [27,28]. The first randomized, double-blind, placebo-controlled trial of ASCs demonstrated their feasibility and safety for myocardial regeneration in 13 patients (ASCs: n = 9; placebo: n = 4) who had successfully recovered from acute myocardial infarction: six months after intracoronary infusion of ASCs or placebo, scar sizes and perfusion defects were significantly improved in the ASC-treatment group (scar size: from 31.6 ± 5.3% to 15.3 ± 2.6%, *p* = 0.002; perfusion defect: from 16.9 ± 2.1% to 10.9 ± 2.4%, *p* = 0.004) but not in the placebo group (scar size: 24.7 ± 9.2% to 24.7 ± 4.1%, *p* = 0.48; perfusion defect: 15.0 ± 4.9% to 16.8 ± 4.3%); LVEF also improved (but not significantly) by ~4% in ASC-treated patients, compared to a 1.7% decline in patients who received placebo [6]. However, subsequent phase 1/2 trials found little evidence of cardiac functional improvement in patients with ischemic cardiomyopathy or refractory chronic myocardial ischemia with left ventricular dysfunction [29,30,31], although the treatment was generally safe and associated with increases in maximal oxygen consumption.

### 2.4. Cardiac Progenitor Cells (CPCs)

Although the myocardium has very little regenerative capacity, several lineages of cardiac cells display progenitor-cell activity. The first putative CPCs were cardiac side population cells, which are activated by cardiac injury and improve cardiac function when transplanted into injured hearts [32]. Furthermore, islet-1—expressing cells differentiate into cardioblasts during embryonic development and persist in the hearts of rats through adulthood [33], while cells that express stem-cell antigen (Sca)-1 participate in the replacement of myocardial cells during normal aging and differentiate into multiple cardiac cell types in response ischemic damage or pressure overload [34].

During culture, cells from cardiac explants self-assemble into spherical aggregates that are enriched for properties of stemness [35], and the cells obtained from these aggregates, cardiosphere-derived cells (CDCs), have been evaluated in two randomized clinical trials. The first trial demonstrated that CDCs were safe and associated with smaller infarct sizes when administered via intracoronary infusion to patients 2 to 4 weeks after myocardial infarction [36]. In the second trial, which was conducted in patients with univentricular heart disease, improvements in ventricular function were significantly greater three months after staged palliation surgery and intracoronary infusion of CDCs than after surgical treatment alone; CDC administration was also associated with significant declines in cardiac fibrosis, and cardiac function remained significantly improved one year after CDC administration [37]. Despite these promising observations, CPC- and CDC-based therapies may not be suitable for all clinical applications, because autologous cardiac cells can only be obtained via invasive biopsy [3].

### 2.5. Embryonic Stem Cells (ESCs) and Induced Pluripotent Stem Cells (iPSCs)

Embryonic stem cells (ESCs) are derived from the inner cell mass of blastocysts and are capable of differentiating into all three embryonic germ layers (endoderm, mesoderm and ectoderm) [38]. When injected directly into the heart, the cardiac environment alone is sufficient to drive the differentiation of ESCs into cardiac cells [39,40], but since ESCs can self-replicate indefinitely, the risk of teratoma formation is high [41]. Thus, ESCs must be differentiated into CMs and other somatic cell types before transplantation, and obtaining sufficiently large, pure, and mature populations of fully differentiated ESC-derived cells, particularly ESC-derived cardiomyocytes (ESC-CMs), can be challenging [3]. Nevertheless, preclinical studies have shown that ESC-CMs are electrically integrated into the native myocardium and promote heart function without forming teratomas [41,42,43,44], and these encouraging results have led to a phase I clinical trial. Only six patients were enrolled, so the study was not designed to provide meaningful efficacy data, but all patients were symptomatically improved, LVEF increased from 28.5 ± 2.8% to 36.0 ± 5.8%, and there was no evidence of teratoma formation or arrhythmogenic complications [45].

Despite these positive results, clinical translation of ESC-derived cells may be impeded by ethical concerns regarding the use of human embryos, as well as the risk of inflammation and immune rejection [8,46,47,48]. Induced pluripotent stem cells (iPSCs) circumvent these limitations, because they can be generated from the somatic cells of individual patients. iPSCs are usually produced via the overexpression of four transcription factors (c-Myc, octamer-binding transcription factor 3/4 [Oct3/4], Sox2, and Kruppel-like factor 4 [Klf4]) [49] and have been reprogrammed from both murine and human cells [3]. Human iPSCs (hiPSCs) were first generated from dermal fibroblasts [50] and have since been reprogrammed from numerous other cell types, including peripheral-blood cells, which could obviate the need for more invasive biopsies [51]. Several protocols for differentiating hiPSCs into cardiomyocytes (hiPSC-CMs) [52], endothelial cells (hiPSC-ECs), smooth-muscle cells (hiPSC-SMCs) [53], and fibroblasts (hiPSC-FBs) [54] have been established, but the results from high-throughput sequencing analyses suggest that the genomes of iPSC-derived cells may be somewhat unstable; thus, even autologous cells may retain some potential for immunogenicity, and the risk of chromosomal abnormalities [55] or other genome-related adverse events (including malignancy) may not be completely abolished [56]. These residual concerns likely explain why iPSC-derived cells have yet to be investigated in clinical trials [3] and have led to the development of other, potentially safer reprogramming methods that use self-replicating RNA [57], synthetically modified RNAs [58], or synthetic transcription factors (Oct4, MyoD, Sox17, Nanog and Mef2c) [59].

Notably, although ESCs and iPSCs are fundamentally different types of cells, iPSCs were specifically designed to reproduce the characteristics of ESCs as closely as possible. Thus, observations in ESCs can very often be replicated in iPSCs (and vice-versa), so the two cell types will be referred to collectively as pluripotent stem cells (PSCs) throughout the remainder of this review.

## 3. Differentiation of PSCs into Cardiomyocytes

Unlike other cardiac cell types, mature cardiomyocytes are non-proliferative, so ESCs and (especially) iPSCs are a particularly important source of cardiomyocytes for regenerative medicine (Figure 1). The most common method for differentiating hiPSCs into cardiomyocytes begins by growing monolayers of the cells on Matrigel-coated plates, and then treating them with medium (e.g., RPMI/B27 or APEL) containing bone-morphogenic protein 4 (BMP4), Activin A, and WNT3A [52,60]. The purity of the differentiated hiPSC-CMs can be increased by overlaying the cells with a mixture of Matrigel and mTeSR1 medium when the monolayer reaches 90% confluence [61]; however, this method is compatible with only a limited number of hiPSC lines [62], and the cytokines can be costly and must be added at specific stages of differentiation. An analogous approach uses small molecules to alter Wnt signaling: first, CHIR99021, which activates Wnt by inhibiting glycogen synthase kinase (GSK) 3, is combined with insulin deprivation to induce the cardiac mesodermal lineage, and then the Wnt inhibitor IWR1 is added with insulin-containing medium to induce the cardiomyocyte phenotype. Spontaneous beating can typically be observed across ~50% of the monolayer surface by the tenth day after differentiation is initiated, and most of the cells express cardiac troponin T (cTnT) [62].

iPSC-CMs can also be generated via inductive co-culture with visceral endodermal-like (END2) cells in serum-free and insulin-free medium, or by embryoid-body formation [52]. Inductive co-culture is quick and requires few cells, so it is particularly useful for confirming that the iPSCs can undergo cardiac mesodermal specification, but the proportion of cells that develop a cardiomyocyte-like phenotype can be as low as 2–3% [52]. Embryoid bodies are spheroids containing an inner layer of ectoderm-like cells and an outer endodermal layer, and are formed via the partial enzymatic dissociation of iPSCs into colonies of 3–20 cells before differentiation. The size and number of cells in the embryoid bodies can be controlled by forcing aggregation with a centrifuge and low attachment V-microwells, by choosing microwells of a specific size and coating the wells with Matrigel, or by using microcontact printing equipment to inoculate micropatterned Matrigel islands with single cells [52]; however, less than 25% of the cells in embryoid bodies differentiate into beating cardiomyocytes [62].

Many protocols for differentiating human PSCs (hPSCs) into cardiac cells are conducted in medium that contains albumin. However, albumin concentrations can be difficult to control, leading to inconsistencies in batch preparation, and like other animal products, albumin may be associated with risks of pathogen contamination and immunogenicity when the cells are administered to patients. In addition to providing a source of nutrition, albumin functions as an antioxidant during differentiation, which suggests that it could be replaced by chemical antioxidants. S12 medium [63] consists of four antioxidant small-molecules (l-ascorbic acid, Trolox, *N*-acetyl-l-cysteine, and sodium pyruvate) plus l-carnitine, which acts as both an antioxidant and a mitochondrial fatty-acid transporter, as well as sodium selenium, human insulin and transferrin, and compounds that contribute to cell-membrane formation (ethanolamine) and contractile activity (linoleic and linolenic acid), in RPMI1640 medium. When this albumin-free medium was used for small-molecule (CHIR99021, IWR1)–induced hiPSC-CM differentiation, differentiation was 20.9% more efficient and produced a 48.6% greater yield with 57% less variability than when differentiation was performed with B27-supplemented medium. S12 medium was also compatible with all five hiPSC lines tested, and the differentiated cells could be maintained in S12 medium for over 100 days [63].

Although the meager amount of endogenous cardiomyocyte renewal in the hearts of adult mammals is driven primarily by whatever residual proliferative capacity remains in pre-existing cardiomyocytes, a small proportion of cells (<0.01% per year), including cardiomyocytes, are believed to descend from resident cardiac progenitor cells [64]. Thus, since differentiation is guided by both the biochemical and structural components of the extracellular matrix [65], PSC-CM differentiation has been conducted in decellularized cardiac tissue which, at least in theory, mimics the native microenvironment of the cardiogenic niche with maximum fidelity; however, all methods of decellularization lead to some disruption of the ECM’s architecture, surface structure, and composition [66]. PSC-CM differentiation has been conducted by seeding the cells into decellularized whole organs and with tissues from both healthy and diseased hearts [67]; for example, when murine PSCs were cultured with maintenance (i.e., non-differentiating) medium in decellularized ECM from patients with end-stage non-ischemic dilated cardiomyopathy, cardiac lineage commitment significantly increased, as evidenced by the expression of cardiac markers such as alpha myosin heavy polypeptide 6 (*MYH6*), cardiac troponin T2 (*TNNT2*), and NK2 homeobox 5 (*NKX2-5*) [68]. Decellularized matrix has also been used to further promote the cardiomyocyte phenotype in hiPSC-CMs that have been differentiated in-vitro: the electrophysiological response and expression of cardiac ion-channel proteins was greater when iPSC-CMs were cultured in decellularized ventricular matrix than on Matrigel [69].

PSC-CM differentiation can also be induced by targeting eomesodermin (also known as T-box brain protein 2 [*TBR2*]) to manipulate Wnt pathway activity [70] (Figure 2). Eomesodermin is encoded by the gene *EOMES* and is crucially required for cardiac development. In mice, nearly all cardiac cells are descended from cells that express *EOMES* during development, and a protocol analogous to the previously described CHIR99021/IWR1 differentiation procedure failed to induce cardiac specification in *EOMES*-knockout hESCs. These observations led to the discovery of a previously unidentified mutual regulatory loop between *EOMES* and Wnt, and subsequent work demonstrated that three days of induced *EOMES* expression (via a doxycycline-regulated promoter) was sufficient for triggering cardiac mesoderm specification in hESCs. When doxycycline treatment was followed by two days of Wnt inhibition, the cells differentiated into monolayers of beating cardiomyocytes that were largely indistinguishable from hESC-CMs obtained via the standard differentiation protocol; however, the doxycycline dose had to be precisely controlled, because too much or too little Wnt activation during the cardiac specification stage led to the development of other cell types.

Kinase domain receptor 1 (KDR1) is expressed by progenitor cells that beget cardiomyocytes, endothelial cells, and smooth-muscle cells during development [71], and three distinct patterns of expression for KDR and C-KIT (a marker for hemangioblast-derived haematopoietic and vascular progenitor cells) are observed in cells from embryoid bodies of induced hESCs: C-KIT-negative cells with low levels of KDR expression (KDR^low^/C-KIT^−^) and C-KIT-positive cells that express high levels of KDR (KDR^high^/C-KIT^+^) or are KDR-negative (KDR^−^/C-KIT^+^). Of these three cell types, KDR^low^/C-KIT^−^ cells had the greatest cardiomyocyte potential: 40–50% of KDR^low^/C-KIT^−^ cells expressed cTnT after 7–10 days of culture [71]. The long-coding RNA GATA6-AS1 also appears to be a key regulator of cardiomyocyte differentiation [72], because although GATA-AS1-knockout human iPSCs remained pluripotent, their ability to differentiate into cardiomyocytes was inhibited. Notably, when data generated from genome-wide RNA-sequencing analyses performed during four stages of murine and human ESC-CM differentiation (i.e., the ESC, mesoderm, cardiac progenitor, and cardiomyocyte stages) were analyzed via multiscale embedded gene co-expression network analysis (MEGENA), 212 significantly co-expressed gene modules were identified; the results from these analyses will provide a valuable foundation for future investigations that could lead to a more complete understanding of the molecular networks that govern PC-CM differentiation [73].

## 4. Purification of Human iPSC-CMs

### 4.1. Lactate-Based Medium and Glucose Starvation

Unlike many other cell types, cardiomyocytes can use lactate as a substrate for energy production when cultured in a low-glucose environment; thus, one of the most efficient methods for removing unwanted cell types (including undifferentiated or partially differentiated human iPSCs) from a cardiomyocyte population involves culturing them in glucose-depleted, lactate-enriched medium [51,62]. The purity human iPSC-CM populations can be increased to >90% via this selection procedure, and although a substantial proportion of the original cardiomyocyte population may be lost (<50%), the rate of attrition is considerably lower than has been reported for other techniques, such as antibody selection (80%) and, consequently, glucose starvation may be the most suitable purification technique for clinical use and other applications that require a large number of cells. However, prolonged exposure (e.g., >3 consecutive days) to low-glucose conditions can lead to substantial declines in cell viability and contractile activity, so the cells must be periodically returned to glucose-containing medium.

### 4.2. Positive or Negative Selection of Labeled Cells

Magnetic-activated cell sorting (MACS) is a simple, inexpensive, and convenient cell-purification procedure with relatively good specificity and yield. Purification of human iPSC-CM can be performed via both positive and negative selection. Positive selection is conducted by labeling cells with phycoerythrin (PE)-conjugated antibodies against known cardiomyocyte markers (e.g., CD172a or CD106) and then passing the cells through a magnetic column [74], which typically retains more than 90% of labeled cells, and just 1.5% of the retained cells are unlabeled. Negative selection uses microbeads and PE-conjugated antibodies to extract cells that express markers for known contaminants (e.g., CD90 and CD140b for fibroblasts, smooth-muscle cells, and endothelial cells [51]); the depletion rate ranges from 31- to 3500-fold for partially and strongly labelled cells, and more than 70% of unlabelled cells are recovered in the eluent [74]. However, the efficiency of both selection methods is crucially dependent on the surface marker(s) used for cell selection, as well as the specificity of the corresponding antibodies, and could decline when used to purify large numbers of cells.

### 4.3. CRISPR/Cas9-Mediated Integration of a Fluorescent Reporter

Purification of human iPSC-CM has also been performed by using CRISPR/Cas-9 genome editing to insert a fluorescent gene next to the gene for a cardiomyocyte-specific marker, and then isolating cells that express the marker protein via fluorescence activated cell sorting (FACS). Selection efficiency can reach 90% and, unlike MACS, is not influenced by antibody specificity, but it remains dependent on marker selection and can be further compromised via off-target effects from the editing procedure. Efficiency and purity are also greater for lower concentrations of cells, which could increase the time required and reduce viability, particularly when large numbers of cells are needed [75]. Furthermore, the genome editing procedure itself can be time consuming and is less desirable for cells that will be used in clinical applications.

## 5. Confirmation of PSC-CM Identity and Functional Characterization

Routine characterization of putative cardiomyocytes typically begins via analysis of cardiomyocyte-specific markers. Immunofluorescence analysis of the expression of cardiac Troponin T (cTnT), sarcomeric α-actinin, myosin light chain 2 (MLC2v), connexin 43, and N cadherin will identify components of the cardiomyocyte cytoskeleton, and quantitative polymerase chain reaction (qPCR) analyses are useful for quantifying the expression of cardiomyocyte-specific genes, such as *NKX2.5*, *ACTN2*, *MYL2*, and *TNNT2*, which encode homeobox protein Nkx-2.5, alpha-actinin 2, myosin regulatory light chain 2, and cardiac troponin T, respectively. Similar analyses can also be performed to evaluate the maturity of PC-CMs by comparing their mRNA and miRNA profiles to those of fetal and adult hearts [76].

The fundamental functional property of cardiomyocytes is their ability to produce contractile force. However, in-vitro measurements of the contractile activity of individual PSC-CMs can be challenging because the cells are typically immature with only partially aligned cytoskeletons and, consequently, contract in several directions. This complex directionality has been accommodated by seeding the cells onto arrays of elastomeric microposts, with each cell forming attachments to multiple posts, and then monitoring the positions of the post tips (i.e., the point of cell attachment) relative to their bases. Studies conducted with iPSC-CMs indicated that the cells typically formed attachments with 13–20 microposts/cell and generated a contractile force of ~15 nN/cell, with a peak contractile power of 29 fW [77]. Contractile measurements have also been conducted in populations of iPSC-CMs suspended in fibrin and stretched between flexible pillars [78], and in collagen rings positioned around elastomeric microcantilevers [79]; subsequent assessments confirmed that the contractile forces increased and decreased in response to positive and negative inotropic factors, respectively.

Characterization of cardiomyocyte functional properties can also include assessments of the cells’ electrical properties, calcium handling, and gap-junction activity (Table 3). Electrical assessments have been conducted via the ruptured-patch whole-cell voltage-clamp technique: cells are mounted in a bath on a microscope stage, and a micropipette containing a wire electrode and electrolytic solution is sealed to the cell membrane; then, the membrane is ruptured with mild suction, and spontaneous action potentials (APs) are recorded and used to calculate the maximum rate of AP rise (Vmax), AP amplitude, AP duration to various stages of recovery (e.g., 30%, 80%), and the resting membrane potential. Intracellular calcium levels can be monitored via fluorimetry after incubating the cells with a calcium-sensitive fluorescent dye, and gap-junction-mediated intracellular signaling can be demonstrated by loading clusters of cells with a fluorescent dye, bleaching a single cell with a high-intensity laser pulse, and then monitoring the re-entry of unbleached dye from adjacent cells (i.e., the ferric reducing ability of plasma [FRAP] assay). When iPSCs were reprogrammed from peripheral blood mononuclear cells and differentiated into cardiomyocytes, results from this battery of assessments demonstrated that action potentials associated with ventricular, atrial, and nodal cardiomyocytes were observed in 50%, 37.5%, and 12.5% of cells, respectively; that intracellular calcium concentrations fluctuated rhythmically, suggesting that the calcium transients were coupled to the cells’ spontaneous beating activity; and confirmed the presence of functional gap junctions [9].

## 6. Promoting PSC-CM Maturation

PSC-CMs are more accurately described as “cardiomyocyte-like” cells, because they possess many—but not all—of the properties that are unique to cardiomyocytes. Human PSC-CMs also more closely resemble embryonic than adult cardiomyocytes [80,81], which has led researchers to seek methods for improving PSC-CM maturity. Treatment with retinoic acid led to increases in human ESC-CM yield when added 2–4 days after differentiation was initiated and to improvements in maturity (e.g., increases in surface area, sarcomere length, multinucleation, and mitochondrial copy number) when added after the cells had begun beating [82]; however, retinoic acid also appears to direct cardiomyocyte differentiation toward an atrial-like phenotype and may promote the reentry-like conduction disorders associated with atrial arrhythmia [83]. Cardiac fibroblasts, which are the primary producers of cardiac extracellular matrix (ECM) proteins, also improved maturation when included in cultures of human PSC-CMs [84], and for engineered heart tissues, efforts to improve human PSC-CM maturity have led to the development of techniques that recreate the electromechanical properties of native myocardium. Electrical stimulation of human PSC-CMs has been associated with improvements in cell elongation, myofilament organization, action potential duration, and calcium transients, as well as increases in the expression of ion channel (*KIR2.1*, *HCN1*, *SCN5A*, *KV4.3*), calcium handling (*CSQ2*, junctin, triadin, SERCA), structural (*CAV3*, *AMP2*), and contractile (myosin heavy chain, myosin light chain) proteins [85]. Furthermore, some evidence suggests that human iPSC-CM maturation can be induced in vivo, after the cells have been administered to infarcted rat hearts, by applying an MSC-containing patch to the epicardial surface over the site of cell administration. MSCs produce a wide variety of growth factors that promote angiogenesis and cell survival, and when administered alone, the MSC-containing patch was associated with improvements in vascularity, but not cardiac performance. However, whereas hiPSC-CMs in animals treated with the cells alone displayed an immature globular phenotype, hiPSC-CMs in animals co-treated with cells and the MSC-containing patch formed larger, more mature rod-shaped structures that resembled adult cardiomyocytes. Patch administration also increased the expression of markers for cardiomyocyte maturation (*TNNT2*, *MYH6* and *MYH7*) in transplanted hiPSC-CMs, and the combined treatment was associated with significantly better measures of cardiac function [86].

## 7. PSC-CMs for Myocardial Repair

Although the feasibility and potential efficacy of transplanted PSC-CMs has frequently been demonstrated in small-animal models, trials in humans have yet to be initiated. Because the hearts of humans and rodents are fundamentally different, both anatomically and physiologically, studies in large animals, such as non-human primates and pigs, are a crucial intermediate step for the clinical translation of human PSC-CM therapy [87]. One primary concern is that since the absolute size of the infarct is so much larger in patients than in rodents, effective treatments will lead to the formation of large islands of transplanted cells that may not be electromechanically integrated within the patient’s intact myocardial tissue, which could lead to arrhythmogenic complications. Arrhythmias have been reported in primate studies of transplanted human ESC-CMs [44], but not in pigs that were administered as combination of human iPSC-CMs, -ECs, and -SMCs, perhaps because the pigs were treated with 100-fold fewer cardiomyocytes [88].

As with other cell therapies, one of the primary factors believed to limit the effectiveness of transplanted PSC-CMs is the exceptionally small proportion of cells that are retained and survive at the site of administration (i.e., the engraftment rate), which can be as low as 3% in mouse models [89]. The loss of transplanted cells can likely be attributed to two distinct processes: (1) many of the administered cells are cleared from the heart and into the peripheral circulation, and (2) most of those that remain succumb to the cytotoxic environment of the injured heart. However, cardiomyocytes are more likely to be retained when administered as spheroids than as single cells, and cardiomyocyte engraftment increases when the cells are delivered with gelatin hydrogel. Thus, these two processes can be combined by suspending cardiomyocyte spheroids in hydrogel before administration, which significantly increases human iPSC-CM retention [86]. The distribution of cells was also more uniform when the spheroid suspension was delivered via a recently developed, 6-needle injection device, rather than a single needle.

The death of transplanted cells during the first few days after administration is likely attributable to ischemia; thus, human PSC-CMs have frequently been co-administered with pro-angiogenic cytokines [90] or, more recently, microvascular fragments [91], to boost vascularity. The microvessels were obtained via digestion of human adipose tissue, and when co-injected with human iPSC-CMs into infarcted rat hearts, survival of the transplanted cells increased six-fold, and measures of cardiac function were significantly better in animals treated with hiPSC-CMs and microvessels than in animals administered human iPSC-CMs alone or in combination with dissociated endothelial cells. Furthermore, ~60% of the microvessels persisted at the injury site for four weeks after administration, and the microvessels appeared to anastomize with the endogenous vascular network.

Engraftment rates are also significantly greater (though still unsatisfactory) when cells are delivered as a patch of engineered heart tissue [87]; but electromechanical integration with the native myocardium remains a concern, and the optimal anatomical location for patch administration has yet to be identified; mesothelial cells in the epicardium could impair integration of an epicardially administered patch, and endocardial placement could increase the risk of thrombosis or embolization. Furthermore, while the cardiomyocytes in engineered heart tissues can effectively reproduce the contractile activity of the myocardium in two dimensions, the twisting motion of the left-ventricle has yet to be replicated, and more effective methods for promoting vascularity are needed, particularly for patches that are more than a few cell-layers thick [87]. Nevertheless, promising results have been reported in several studies, including at least one conducted with patches large enough (4 cm × 2 cm × 1.25 mm) for clinical use [92]. The patches were composed of human -CMs, -ECs, and -SMCs suspended in fibrin, and when tested in a large-animal (pig) model of myocardial injury, the patches were associated with significant improvements in cardiac function, infarct size, wall stress, and hypertrophy.

Although the reprogramming of autologous somatic cells into human iPSCs may induce genomic changes that could, at least in theory, induce an inflammatory response when the cells are re-administered to a patient [55], the proportion of autologous hiPSC-CMs that are rejected by the patient’s immune system is likely to be minimal. Nevertheless, the time required for the reprogramming and differentiation procedures precludes the use of autologous hiPSC-derived cells for emergency situation, such as acute myocardial infarction. Thus, researchers have attempted to generate a line of “universal donor” hiPSCs [93] by knocking out components of the human leukocyte antigen system (e.g., *HLA-A/B/C* and *CIITA*) to prevent the adaptive immune response, while simultaneously overexpressing immunomodulatory factors (e.g., *PD-L1*, *HLA-G*, and *CD47*) so that the cells can escape detection by the innate immune system. Despite substantial progress, some evidence suggests that these cells can still induce an immune response in vivo and, consequently, a more complete understanding of their immunoreactivity is needed before a truly “off-the-shelf” hiPSC-based cell therapy can be developed. Immune-rejection could also be minimized via HLA typing and matching protocols analogous to those currently conducted for patients undergoing organ transplantation [94], provided the supply hiPSC-derived cells becomes suitably large and immunologically diverse.

## 8. Direct Transdifferentiation of Somatic Cells into Cardiomyocytes

Somatic cells can also be transdifferentiated directly into cardiomyocytes (i.e., induced cardiomyocyte-like cells (iCMLCs)) (Figure 1b), without first being reprogrammed into PSCs, via the delivery of transcription factors, microRNAs, and small molecules [3,4,5,6,7,8,9] (Table 2). However, the efficiency of transdifferentiation is likely to depend on the lineage of the somatic cell (e.g., cardiomyocytes may be more effectively generated from cardiac fibroblasts than from fibroblasts in other organs) and current protocols typically yield only a small percentage of cells that display recognizably cardiomyocyte-like properties [95]. Human dermal fibroblasts have been transdifferentiated into iCMLCs via the overexpression of five cardiac transcription factors (*GATA4*, *TBX5*, *MEF2C*, *MYOCD*, *NKX2-5*; i.e., the GTMMN protocol (Table 2 and Table 3) in low-serum medium supplemented with a Janus kinase inhibitor (JAKi) [9,51,52,63,95,96,97]. The cardiomyocyte phenotype was evaluated via the expression of alpha-actinin 2 (*ACTN2*) and cardiac muscle troponin T (*TNNT2*); however, only 0.21% of the treated cells expressed both markers. Transdifferentiation efficiency increased to 3.8% when a single dose of *miR-1* and miR-133a was administered before a continuous, two-week period of GTMMN induction, but the cytoskeletal organization of *ACTN2* and *TNNT2* was poorly developed, and the transdifferentiated cells did not spontaneously contract, despite evidence of functional calcium channels. Similar observations have been reported when human dermal fibroblasts were transdifferentiated with other combinations of transcription factors (e.g., *GATA4*, *MEF2C*, *TBX5*, *ESRRG*, *MESP1*, *MYOCD*, and *ZFPM2* [98]; *GATA4*, *MEF2C*, *TBX5*, *MESP1*, and *MYOCD* [97]) or transcription factors and microRNAs (*GATA4*, *TBX5*, *HAND2*, *MYOCD*, *miR-1*, and *miR-133* [99]): none of the cells displayed spontaneous contractile activity [96].

iCMLCs have also been generated via treatment with small molecules (i.e., small-molecule transdifferentiation [SMT]), and unlike cells transdifferentiated via the overexpression of transcription factors (with or without concomitant miRNA administration), the iCMLCs generated via SMT uniformly displayed contractile properties [100]. Human foreskin fibroblasts were treated for 6 days with a combination of seven compounds (CHIR99021, A83-01, BIX01294, AS8351, SC1, Y27632, OAC2) and then cultured in medium containing activin A, BMP4, vascular endothelial growth factor, and CHIR99021 for five days, which induced the formation of clusters of beating cells, and the number of beating clusters was further increased by the inclusion of two more compounds (SU16F and JNJ10198406) that inhibit platelet-derived growth factor signaling [100]. The same combination of nine compounds was also used to generate iCMLCs from human fetal lung fibroblasts, and assessments of cellular structure (e.g., patterns of alpha-smooth muscle and ventricular myosin light-chain 2v expression), action potentials, and calcium transients in the SMT-iCMLCs cells were largely indistinguishable from those in hiPSC-CMs. Thus, despite the relatively low yield (only ~6% of fibroblasts treated with the nine SMT compounds expressed cTnT), SMT-iCMLCs may be preferable to hiPSC-CMs, because the transdifferentiation procedure does not disrupt genomic stability, though whether equivalent results can be obtained with fibroblasts from adult patients, rather than fetal or neonatal tissues, has yet to be determined. Notably, when the SMT-iCMLCs were injected into infarcted mouse hearts, the cells continued to express cardiomyocyte markers and display well-organized sarcomeres two weeks later, which confirmed that the cardiomyocyte phenotype of SMT-iCMLCs remained stable after administration.

The success of in-vitro methods for transdifferentiating fibroblasts into iCMLCs suggests that endogenous cardiac fibroblasts could be an in-situ source of cardiomyocytes for myocardial regeneration. In infarcted mouse hearts, the in-vivo transdifferentiation of cardiac fibroblasts into iCMLCs has been induced via retroviral overexpression of three (*GATA4*, *MEF2C*, *TBX5* [GMT]) or four (GMT plus *HAND* [GHMT]) transcription factors [95]. GMT-induced iCMLCs were binuclear, developed sarcomeres, and displayed evidence of cardiomyocyte-like gene expression, action potentials, and electrical coupling [102]; and the treatment was associated with improvements in infarct size and cardiac function that were further enhanced when thymosin β4 was co-administered to promote the migration of fibroblasts to the injury site. iCMLCs can also likely be generated in vivo via SMT, which is less costly than transcription-factor-mediated transdifferentiation and can be administered with better temporal control and less risk of immunogenicity; however, small molecules are more likely to enter the peripheral circulation and, consequently, may be associated with off-target adverse effects. Thus, future applications of both transcriptional and small-molecule in-situ iCMLC transdifferentiation will require the development of more efficient and precise delivery methods.

## 9. Recent Clinical Trials

Clinical trials of stem-cell therapy for heart failure are reasonably numerous, but most have very few participants. A relatively large phase II randomized double-blind study, Combination of Mesenchymal and C-kit+ Cardiac Stem Cells as Regenerative Therapy for Heart Failure (CONCERT-HF) [103], enrolled 144 patients (125 of whom completed the study) with ischemic cardiac injury and New York Heart Association (NYHA) class I-III symptoms of heart failure. Patients received 150 million autologous BM MSCs, 5 million c-kit+ CPCs, or both via transendocardial injection into the left ventricle, and the placebo group was administered PlasmaLyte A. Despite demonstrating safety of treatment, experiments failed to show significant improvements in measures of cardiac function: at the end of the 6-month follow-up period, differences in LVEF between the MSC, CPC, or MSC+CPC group and placebo were 0.70 ± 1.93 (*p* = 0.499), 1.38 ± 2.11 (*p* = 0.578), and −0.08 ± 2.11 (*p* = 0.993), respectively, and analyses of other primary outcome measures, such as ventricular wall-strain and ischemic scar size, were also inconclusive. However, the occurrence of major adverse cardiac events was significantly decreased (from 28% to 6.5%) in the CPC group (*p* = 0.043), and quality of life scores were significantly better in patients treated with MSCs alone (*p* = 0.050) or both MSCs and CPCs (*p* = 0.023) than in placebo-treated patients [103]. Notably, despite the relatively large total number of study participants, the number of patients in each treatment arm was still small (29–33), which likely limited the power of the study to detect significant differences between groups, and the study evoked the failure of cell growth in certain patients that could have negatively impacted outcomes in the patients receiving autologous BM-MSCs.

Ongoing clinical trials include a Danish multicenter phase II, double-blind, randomized, placebo-controlled study [104] of allogeneic ASC in 81 patients with chronic ischemic heart disease, LVEF below 45%, and NYHA class II-III symptoms, and the University of Campania Luigi Vanvitelli is currently recruiting patients with refractory heart failure for a phase IV study of intra-cardiac stem cell infusion [105]. Furthermore, whereas most clinical trials have investigated just a single dose of cell therapy, researchers at Shanghai East Hospital propose to study a three-dose regimen of umbilical cord MSCs in a randomized, double-blind, placebo-controlled, phase I explorative trial; the cells will be intravenously infused at 6-week intervals in 40 patients with reduced LVEF (<40%) and NYHA class II-IV symptoms [106].

The first clinical use of hiPSC-CMs was announced by researchers at Osaka University in January 2020 [107]. The physician-initiated study is planned as a phase 1 trial and will evaluate the safety and potential efficacy of transplanted hiPSC-CM sheets in the hearts of 10 patients with chronic ischemic cardiomyopathy and associated symptoms. Notably, the sheets are believed to release growth factors that help regenerate the damaged muscle, but are not expected to integrate into the native heart tissue, and because the cells are allogeneic, the study will also assesses the safety and tolerability of immunosuppressants in this patient population [108,109].

## 10. Conclusions

The principal objective of regenerative myocardial therapy is to remuscularize the site of infarction with functional contractile tissue, which necessarily requires large numbers of cardiomyocytes. However, the cardiomyocytes of adult hearts are non-proliferative, so successful treatments will likely rely on the development of techniques for generating cardiomyocytes from other cellular sources. Numerous varieties of somatic stem/progenitor cells have been investigated with varying levels of success, but iPSCs have emerged as perhaps the most promising, because their multipotency and capacity for self-renewal are (at least in theory) unlimited, and since they can be reprogrammed from a patient’s own somatic cells, they are unlikely to induce an immune response after administration. The hiPSC-CM technology could also stimulate progress in the development of new pharmacological treatments by providing a reliable supply of cardiomyocytes for in-vitro drug testing. However, clinical studies of iPSC-CMs CMs have only started to emerge, in part, because of residual concerns about genomic instability, and although the delivery of both dissociated hiPSC-CMs and hiPSC-CM-containing engineered heart-tissue patches has improved myocardial performance and infarct size in small- and large-animal models, engraftment rates remain unsatisfactory, and arrhythmogenic complications have not been completely abolished. Furthermore, the (very limited) success of attempts to transdifferentiate fibroblasts directly into iCMLCs suggests that the pool of cardiac fibroblasts could also be a viable source of cardiomyocytes for myocardial repair. Continued development of these novel and cutting-edge cardiac therapies will be crucial as the prevalence of heart failure continues to rise in the US and elsewhere and will likely be exacerbated as patients recover from the acute cardiovascular complications associated with the ongoing COVID-19 pandemic.

## Figures and Tables

**Figure 1 cells-10-03125-f001:**
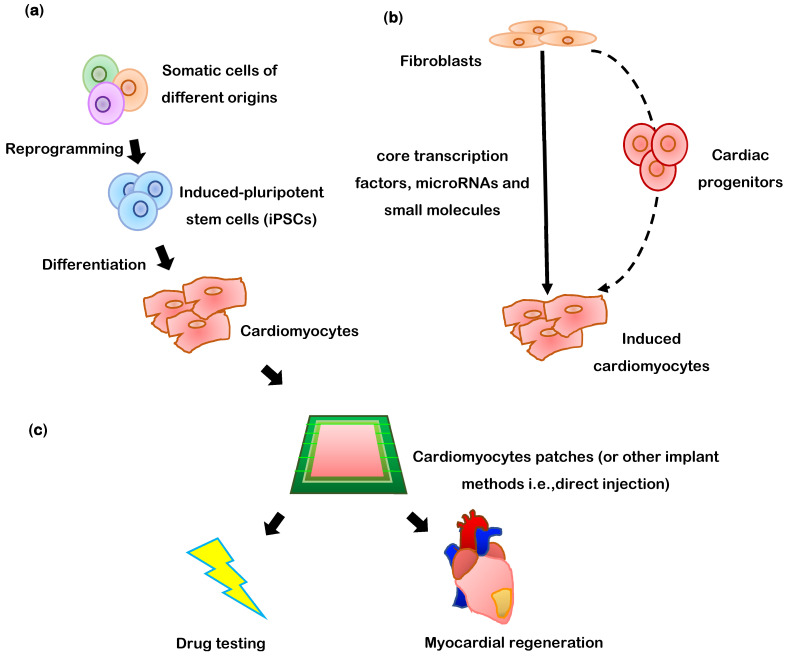
The two principal methods for generating cardiomyocytes. (**a**) In-Vitro: Somatic cells are reprogrammed into iPSCs, then, the iPSCs are differentiated into cardiomyocyte-like cells which can be directly injected into infarcted myocardium or assembled into a patch of engineered cardiac tissue for therapeutic implantation or drug testing. (**b**) In-Vivo: One specific type of somatic cell (typically fibroblasts) is treated with targeted factors to induce transdifferentiation into cardiomyocyte-like cells. (**c**) Generated cardiomyocytes can be delivered as cardiomyocyte patches into damaged heart tissue, or used for drug testing.

**Figure 2 cells-10-03125-f002:**
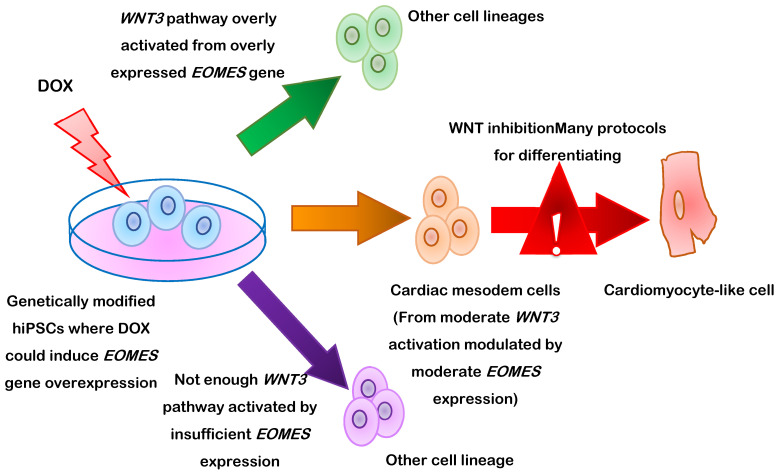
Schematic representation of human induced pluripotent stem cell (hiPSC-CM) differentiation via the manipulation of eomesodermin (*EOMES*) gene expression via a doxycycline (DOX)-regulated promoter.

**Table 1 cells-10-03125-t001:** Types of cells used in cardiomyocyte (CM) regeneration [3,4,5,6,7,8,9].

Cell Types	Skeletal Myoblasts	Bone Marrow-Derived Hematopoietic Stem Cells (Bm-Hscs)	Bone Marrow-Derived Endothelial Progenitor Cells (Bm-Epcs)	Bone Marrow-Derived Mesenchymal Stem Cells (Bm-Mscs)
origin	Autologous muscle biopsies (easy)	Autologous bone marrow/blood (easy)	Autologous bone marrow/blood (easy)	Autologous tissues (easy)
ethical concerns	Low	Low	Low	Low
tumorigenicity risk	Low	Low	Low	Low
cell quantity	Sufficient	Limited	Limited	Limited
differentiation potentials into cms	Cannot generate functional CMs	Limited potentials	Limited potentials	Limited potentials
growth	Rapid in vitro expansion	Rapid in vitro expansion	Rapid in vitro expansion	Rapid in vitro expansion
	Resist ischemic conditions	Heterogenous cell population	Heterogenous cell population	Heterogenous cell population
immunologic rejection risks	Low	Low	Low	Low
other advantages	-	Proved safe in clinical trials	Proved safe in clinical trials	-
	-	Promote vasculogenesis Therapeutic secretome		-
other inconveniences	Ventricular arrhythmia hazard	Encourage inflammation		Ambiguous therapeutic results
**Cell Types**	**Adipose-Derived Stem Cells (ASCS)**	**Cardiac Stem Cells (cscs) and Cardiac Progenitor Cells (CPCS)**	**Embryonic Stem Cells (ESC)**	**Induced Pluripotent Stem Cell (IPSC)**
origin	Autologous tissues (easy)	Autologous myocardial biopsies (invasive)	Inner cell mass of blastocysts from in vitro fecundation (non-autologous)	Reprogrammed from autologous cells (easy access)
ethical concerns	Low	Low	High	Low
tumorigenicity risk	Low	Low	High	High
cell quantity	Sufficient	Limited	Unlimited	Unlimited
differentiation potentials into cms	Limited potentials	Ambiguous results	Pluripotent differentiation potentials Generate CMs capable of integrating electromagnetically into the host myocardium	Pluripotent differentiation potentials Generate CMs capable of integrating electromagnetically into the host myocardium
growth	Rapid in vitro expansion	Insufficient cell characterization as CMs	Difficult to generate pure and mature cardiomyocytes in large quantities	Difficult to generate pure and mature cardiomyocytes in large quantities
	Heterogenous cell population	Heterogenous cell population	Unavailability	Lack of standardized generation
				Low induction efficiency
immunologic rejection risks	Low	Low	High risks require immunosuppression (non-autologous)	Low
other advantages	-	Proved safe in clinical trials	-	-
	-	-	-	-
other inconveniences	-	-	-	-

**Table 2 cells-10-03125-t002:** Factors used for transdifferentiating somatic cells directly into cardiomyocytes [95,96,100,101].

Original Cell	Dermal Fibroblast (DF)	Human Cardiac Fibroblast (HCF)	Embryonic Stem Cell (esc), Fetal Heart (FH), Neonatal Skin	Human Cardiac Fibroblast (HCF)	Human Foreskin Fibroblast (HFF)	Human Foreskin Fibroblast (HFF)
factors	ETS2	*GATA4*	*GATA4*	*GATA4*	*GATA4*	CHIR99021
	*MESP1*	*MEF2C*	*MEF2C*	*MEF2C*	*HAND2*	A83-0
		*TBX5*	*TBX5*	*TBX5*	*TBX5*	BIX01294
		*MESP1*	*ESRRG*	*MESP1*	*MYOCD*	AS8351
		*MYOCD*	*MESP1*	*MYOCD*	*miR-1*	SC1
			*MYOCD*	*miR-133*	*miR-133*	Y27632
			ZFPM2			OAC2
						SU16F
						JNJ10198409
markers (efficiency)	NKX2.5-tdTomato^+^ (30 colonies/plate of cardiac progenitor)	cTnT^+^ (5.9%)	α-MHC-mCherry^+^ (15.8%)	cTnT^+^ (27.8%)	cTnT^+^ (34.1%)	cTnT^+^ (6.6%)
		α-actinin^+^ (5.5%)	α-MHC-mCherry^+^ & cTnT^+^ (13%)	α-actinin^+^ (8%)		
action potential	Negative	Positive	Positive	Not detected	Not detected	Positive
Ca^2+^ transient	Negative	Positive	Positive	Positive	Positive	Positive
beating	Negative	Positive	Not detected	Positive	Positive	Positive

ETS2 (V-ets Erythroblastosis virus E26 oncogene homolog 2), *MESP1* (Mesoderm posterior BHLH transcription factor 1), *GATA4* (GATA-binding protein 4), *MEF2C* (Myocyte-specific enhancer factor 2C), *TBX5* (T-box transcription factor 5), *MYOCD* (Myocardin), *ESRRG* (Estrogen-related receptor gamma), ZFPM2 (Zinc finger protein, FOG family member 2), *miR-133* (Micro-RNA-133), *HAND2* (Heart and neural crest derivatives-expressed protein 2), *miR-1* (Micro-RNA-1), CHIR99021 (6-((2-((4-(2,4-Dichlorophenyl)-5-(4-methyl-1H-imidazol-2-yl)pyrimidin-2-yl)amino)ethyl)amino)nicotinonitrile, a chemical compound), A83-0 (3-(6-Methyl-2-pyridinyl)-*N*-phenyl-4-(4-quinolinyl)-1H-pyrazole-1-carbothioamide, a potent inhibitor of TGF-β type I receptor ALK5 kinase), BIX01294 (quinazoline derivate, an inhibitor of a G9a histone methyltransferase), AS8351 (311 Iron chelator, a histone demethylase inhibitor), SC1 (Pluripotin), Y27632 ((1R,4r)-4-((R)-1-aminoethyl)-*N*-(pyridin-4-yl)cyclohexanecarboxamide, a selective inhibitor of p160ROCK (rho-associated protein kinase)), OAC2 (*N*-1H-Indol-5-yl-benzamide), SU16F (5-[1,2-Dihydro-2-oxo-6-phenyl-3H-indol-3-ylidene)methyl]-2,4-dimethyl-1H-pyrrole-3-propanoic acid, a potent and selective PDGFRβ inhibitor), JNJ10198409 (3-Fluoro-*N*-(6,7-dimethoxy-2,4-dihydroindeno[1,2-c]pyrazol-3-yl)phenylamine, *N*-(3-fluorophenyl)-2,4-dihydro-6,7-dimethoxy-Indeno[1,2-c]pyrazol-3-amine), NKX2.5-tdTomato (NK2 homeobox 5 protein coupled with tdtomato red fluorescent protein), cTnT (Cardiac troponin T), a-MHC-mCherry (alpha major histocompatibility complex coupled with mCherry red fluorescent protein).

**Table 3 cells-10-03125-t003:** Characteristics of adult cardiomyocytes (CMs), iPSC-CMs and cardiomyocytes transdifferentiated from somatic cells. [9,51,52,63,95,97].

Cells Type	Adult Cms	Ipsc-Cms	Transdifferentiated Cms
differentiation efficiency	-	>80%	~60% expressing cTnT^+^ and α-actinin^+^ markers
size	Membrane capacitance 150 pF	Small size (membrane capacitance 18 pF), 1/10 size of adult CMs	Small size
nucleus	Bi- or multi-nuclear	Mononuclear	Mononuclear
morphology	Rod-shape	Circular shape Irregular shape	Spindle-shape
sarcomere	Highly organized	Better organized	Disarrayed
primary metabolic substrate	Fatty acid	Glucose	Glucose
markers	α-MHC^+^	α-MHC^+^	α-MHC^+^
	α-actinin^+^	α-actinin^+^	α-actinin^+^
	Troponin T^+^	Troponin T^+^	Troponin T^+^
Ca^2+^ transient	Positive	Positive	Positive (few induced CMs)
electrophysiology	Resting membrane potential −90 mV (quicker action potential)	Resting membrane potential −60 mV (slower action potential)	Resting membrane potential −48 mV (slowest action potential)

cTnT (Cardiac troponin T), α-MHC (alpha major histocompatibility complex).

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
