# Peer review of "Regenerating Damaged Myocardium: A Review of Stem-Cell Therapies for Heart Failure"

_cells, 2021, doi:10.3390/cells10113125_

Round 1

Reviewer 1 Report

This is a very comprehensive review for the field of stem cell therapy and other regenerative approaches. 

I think the authors need to emphasize on the limitations of the stem cell therapy and how could we address these limitations and what is the futurs directions.

Author Response

REVIEW 1: Comments and Suggestions for Authors

This is a very comprehensive review for the field of stem cell therapy and other regenerative approaches. I think the authors need to emphasize on the limitations of the stem cell therapy and how could we address these limitations and what is the futures directions.

Response: We appreciate this comment. The use of stem-cell therapy for treatment of myocardial disease is limited primarily by low rates of engraftment (discussed in lines 426-438 of our manuscript), concerns about the maturity (lines 408-412) and potential arrhythmogenicity (lines 420-425) of transplanted cardiomyocytes, and questions about the potential genomic instability of iPSC-derived cells (lines 465-467). We also acknowledge that the time required to reprogram somatic cells into hiPSCs and then differentiate them into cardiomyocytes and other cell types before administration precludes the use of autologous iPSC-derived cells for emergency conditions, such as acute myocardial infarction (lines 469-471). However, we have added a brief discussion of the first clinical trial of iPSC-CM administration (lines 584-590), which will be conducted with allogeneic cells and, consequently, will provide clues about the feasibility of allogeneic iPSC-derived cell administration.

Reviewer 2 Report

Response.

The review is dedicated to review pluripotent stem cell types and other stem cells role in cardiomyogenic regeneration. The execution of review is definitely not the best. It is a mixture of information, too much of everything, without clear structure and depth analysis of information. The review needs a very strong rewriting and consideration.  

  1. The abstract says that review will be focused mainly to the iPSC, while on the line 50 – is written that will be focusing primarily on embryonic stem cells (ESCs) and iPSCs and then a lot of information in the text and tables about the other types of the stem cells.
  2. If the review is focusing on iPSC and ESC, the review should start from these cells, not from other types.
  3. Table 1 as well as tables 2 and 3 do not have any citations. Each statement should be cited showing where it is taken from. Table 2 has a column title with the mixture of cell types “EMBRYONIC STEM CELL (ESC), FETAL HEART (FH), NEONATAL SKIN”, which is confusing. How possible to make an information mixture from different cell types especially without any citations?
  4. The Table 1 does not confirm that review is focusing primarily on embryonic stem cells (ESCs) and iPSCs. It is rather comparable information.
  5. Only few lines are written about the skeletal myoblasts.
  6. The citation about application of HSC in cardiac regeneration was not presented at all (line 90).
  7. Adipose-derived stem cells (ASCs) are presented as special type of stem cells (in Table 1 as well). Usually these cells are named as adipose-derived MSC. If ASC are different from the adipose-derived MSC or other types of MSC, it should be exactly stated and markers should be shown.
  8. Line 182. The “(especially)” does not need parentheses.
  9. Line 183. Figure 1 is wrongly addressed in the text, i.e. the text is about ESC and iPSC, but the figure is about cell-based cardiac technologies in vitro and in vivo. The Fig. 1 is not complete, other differentiation types are missing.
  10. 1 – the statement “Chemical factor (core transcription factors, microRNAs and…..) is not finished. Additionally, chemical factors are usually named chemical compounds, not RNA or transcription factors.
  11. The part: „Differentiation of PSCs into cardiomyocytes“ is the worst – differentiation protocols, ECM-based techniques, iPSC and ESC, cardiac development and many other facts are mixed up into one chapter. iPSC and ESC are not the same, they just belong to the PSC group, so, to talk about these cells as about one type PSC is not correct. Further chapters also have a PSC-related information, which should be separated into ESC and iPSC.
  12. Very often in the review there is no information about the source of models or cells the authors are talking about, i.e. are tehy human or mouse or other animals’. For example, line 350 or lines 375-378 or 406-410: what the authors mean saying “hPSC-CMs“ or “PSC-CMs”, from what type of pluripotent stem cell the CMs were generated?
  13. The review has no clear structure, too much of information, which is very often superficial.
  14. Chapters 5 and 6 is also made as mixture of CM maturation information.
  15. Chapter 7 is not only about the PSC cells, but is also about the engineering techniques.

Author Response

REVIEW 2: Comments and Suggestions for Authors

The review is dedicated to review pluripotent stem cell types and other stem cells role in cardiomyogenic regeneration. The execution of review is definitely not the best. It is a mixture of information, too much of everything, without clear structure and depth analysis of information. The review needs a very strong rewriting and consideration.

  1. The abstract says that review will be focused mainly to the iPSC, while on the line 50 – is written that will be focusing primarily on embryonic stem cells (ESCs) and iPSCs and then a lot of information in the text and tables about the other types of the stem cells.

Response: We have changed the indicated portion in the Abstract (lines 10-20) to read “This review begins by briefly discussing a variety of somatic stem- and progenitor-cell populations that were frequently studied in early investigations of regenerative myocardial therapy and then focuses primarily on pluripotent stem cells (PSCs), especially induced-pluripotent stem cells (iPSCs), which have emerged as perhaps the most promising source of cardiomyocytes for both therapeutic applications and drug testing.”

  1. If the review is focusing on iPSC and ESC, the review should start from these cells, not from other types.

Response: As noted in our response to the previous comment, much of the early work in cell-based regenerative myocardial therapy was conducted with somatic stem-cell populations; thus, we discuss them first to provide a historical background, before focusing on PSC-derived cells. However, we recognize that our inclusion of ESCs and iPSCs under the subheading “Somatic stem and progenitor cells” was incorrect, so they are discussed under a separate heading in our revised manuscript.

  1. Table 1 as well as tables 2 and 3 do not have any citations. Each statement should be cited showing where it is taken from. Table 2 has a column title with the mixture of cell types “EMBRYONIC STEM CELL (ESC), FETAL HEART (FH), NEONATAL SKIN”, which is confusing. How possible to make an information mixture from different cell types especially without any citations?

Response: The citations in the tables appear to have been lost during formatting. We apologize for this error and have re-inserted them.

  1. The Table 1 does not confirm that review is focusing primarily on embryonic stem cells (ESCs) and iPSCs. It is rather comparable information.

Response: We agree. Table 1 summarizes some of the key differences among various stem/progenitor cell populations, which will enable the reader to understand why PSCs, and especially iPSCs, are generally considered to be superior for cardiomyocyte generation.

  1. Only few lines are written about the skeletal myoblasts.

Response: Skeletal myoblasts were frequently used in early preclinical studies of regenerative myocardial therapy; however, no clinical trials have been completed because (as noted in lines 62-65 of our manuscript) “… transplanted skeletal myoblasts have been associated with a high risk of ventricular arrhythmia...” Thus, since the development of skeletal myoblasts for therapeutic purposes appears to have stalled, we do not believe it is appropriate to include a lengthy discussion of them in this manuscript. But thank you for pointing it out.

  1. The citation about application of HSC in cardiac regeneration was not presented at all (line 90).

Response: In lines 78-80 of our manuscript, we state that “EPCs are a provasculogenic subpopulation of HSCs that express CD133 and other lineage markers…”; then, we discuss the results of a clinical trial of CD133+ cells in patients who underwent coronary artery bypass graft surgery (Yerebakan, et al., 2011). Thus, the cells used in Yerebakan, et al., were HSCs, but we believe it is more appropriate to maintain the notation used by the authors of the original study.

  1. Adipose-derived stem cells (ASCs) are presented as special type of stem cells (in Table 1 as well). Usually these cells are named as adipose-derived MSC. If ASC are different from the adipose-derived MSC or other types of MSC, it should be exactly stated and markers should be shown.

Response: We have addressed this comment by including the following statement in our revised manuscript (lines 105-108): “Nevertheless, although ASCs are sometimes referred to as ‘adipose-derived MSCs,’ they are fundamentally different from the MSCs present in other tissues; for example, BM-derived MSCs express the surface marker CD146, while ASCs do not.”

  1. Line 182. The “(especially)” does not need parentheses.

Response: We have incorporated this comment as requested.

  1. Line 183. Figure 1 is wrongly addressed in the text, i.e. the text is about ESC and iPSC, but the figure is about cell-based cardiac technologies in vitro and in vivo. The Fig. 1 is not complete, other differentiation types are missing.

Response: Figure 1 illustrates the two of the most promising methods for restoring the cardiomyocytes that are lost in response to myocardial injury or disease: 1) in-vitro reprogramming of somatic cells into iPSCs, which are then differentiated into cardiomyocytes, and 2) in-situ reprogramming of cardiac fibroblasts into cardiomyocytes. However, we acknowledge that these two methods are discussed in separate sections of the manuscript, so in our revised submission, we have called out each of the two panels individually (i.e., Figure 1a in the section about iPSCs and Figure 1b in the section about in-situ reprogramming).

  1. the statement “Chemical factor (core transcription factors, microRNAs and…..) is not finished. Additionally, chemical factors are usually named chemical compounds, not RNA or transcription factors.

Response: The remainder of the text box was hidden when the image was resized. We apologize for this error and have corrected it in our revision. We have also removed the word “chemical” from the corresponding portion of the figure.

  1. The part: „Differentiation of PSCs into cardiomyocytes“ is the worst – differentiation protocols, ECM-based techniques, iPSC and ESC, cardiac development and many other facts are mixed up into one chapter. iPSC and ESC are not the same, they just belong to the PSC group, so, to talk about these cells as about one type PSC is not correct. Further chapters also have a PSC-related information, which should be separated into ESC and iPSC.

Response: Thank you. The purpose of this section is to summarize the methods used to differentiate PSCs into cardiomyocytes. Thus, we begin with a paragraph about the most common protocols (i.e., culturing on Matrigel-coated plates with medium containing BMP4, Activin A, and WNT3A), before discussing other methods (e.g., co-culture with END2 cells, embryoid-body formation, the use of decellularized cardiac tissue, targeting eomesodermin) and considerations (e.g., albumin as a nutrition source and antioxidant, balancing the activation of Wnt signaling). Furthermore, although we acknowledge that iPSC and ESCs are not identical, both cell types are likely compatible with the same differentiation protocols, because (as noted in lines 157-160 of our manuscript): “… iPSCs were specifically designed to reproduce the characteristics of ESCs as closely as possible. Thus, observations in ESCs can very often be replicated in iPSCs (and vice-versa),…”

  1. Very often in the review there is no information about the source of models or cells the authors are talking about, i.e. are tehy human or mouse or other animals’. For example, line 350 or lines 375-378 or 406-410: what the authors mean saying “hPSC-CMs“ or “PSC-CMs”, from what type of pluripotent stem cell the CMs were generated?

Response: We apologize for the confusion and have revised our manuscript accordingly. As noted previously, much of what can be observed in iPSCs and ESCs is similar, so we have used the term PSC for general observations that are applicable to both cell types, and the more specific terms (e.g., hiPSC, hESC) when discussing observations related to each individual cell type.

  1. The review has no clear structure, too much of information, which is very often superficial.

Response: We have revised the manuscript to improve clarity.

  1. Chapters 5 and 6 is also made as mixture of CM maturation information.

Response: Our discussion of methods for promoting PSC-CM maturation is included in a single paragraph under its own subheading. As noted previously, observations in iPSC- and ESC-CMs are largely similar, so the two cell types are discussed concurrently.

  1. Chapter 7 is not only about the PSC cells, but is also about the engineering techniques.

Response: We have included a paragraph on engineered tissues because, as noted in lines 450-452 and 474-476 (respectively) of our manuscript: “… one of the primary factors believed to limit the effectiveness of transplanted PSC-CMs is the exceptionally small… the engraftment rate” and “Engraftment rates are also significantly greater… when cells are delivered as a patch of engineered heart tissue.” However, we do not discuss the methods and techniques used to generate engineered cardiac tissues, because we believe they are beyond the scope of this review.

Reviewer 3 Report

The review manuscript entitled “Regenerating damaged myocardium: a review of stem-cell therapies for heart failure” by Fan D. et al is well written review and provides a comprehensive overview of stem cell-based cell replacement therapies of heart failure. There a few concerns listed below that need to be addressed by the  authors in this manuscript before its acceptance for publication:

  1. The authors should consider including appropriate references citing each of the observations in the table 1 and 2
  2. In table 1, the authors’ statements as “Unavailability” and “Lack of standardized generation” under the Growth section (page # 3, second row/line) for ESC and IPSC respectively are not clear and look more misleading. The authors should clarify on these controversial terms.
  3. The authors have not defined “PC-CM” mentioned on page #8, line # 286 and page #9, line # 301.
  4. More mature atrial/ventricular cells don’t exhibit spontaneous contractility, much more like the native adult cardiomyocytes. Authors state that “However, declines in 299 spontaneous contractile activity can occur if the conditions are maintained for more than 300 3 consecutive days, perhaps because PC-CMs are less mature than primary adult cardio- 301 myocytes, so the cells must be periodically returned to glucose-containing medium.” Since it is normally anticipated that more mature cardiomyocytes would lose spontaneous contractile activity, the authors’ above statement is misleading. The authors should clarify on this discrepancy with appropriate scientific evidence (specifically, why the cells should be returned to glucose-containing medium?)

  1. The authors mention in the abstract that this review manuscript will focus primarily on iPSCs in the context of cell therapy for heart failure. But, the authors have not included the currently ongoing clinical trials with human iPSC-cardiomyocytes, while they cited many clinical trials with other stem cell types. If the authors could cover current status of the iPSC-CM based clinical trials, that will greatly strengthen this review manuscript.

Author Response

REVIEW 3: Comments and Suggestions for Authors

The review manuscript entitled “Regenerating damaged myocardium: a review of stem-cell therapies for heart failure” by Fan D. et al is well written review and provides a comprehensive overview of stem cell-based cell replacement therapies of heart failure. There a few concerns listed below that need to be addressed by the authors in this manuscript before its acceptance for publication:

  1. The authors should consider including appropriate references citing each of the observations in the table 1 and 2

Response: The citations in the tables appear to have been lost during formatting. We apologize for this error and have re-inserted them.

  1. In table 1, the authors’ statements as “Unavailability” and “Lack of standardized generation” under the Growth section (page # 3, second row/line) for ESC and IPSC respectively are not clear and look more misleading. The authors should clarify on these controversial terms.

Response: We thank the reviewer for bringing this issue to our attention. We have replaced the indicated terms with “lack of data,” which is more accurate.

  1. The authors have not defined “PC-CM” mentioned on page #8, line # 286 and page #9, line # 301.

Response: We apologize for this typographical error, the correct term is PSC-CM, which has been defined previously in the manuscript.

  1. More mature atrial/ventricular cells don’t exhibit spontaneous contractility, much more like the native adult cardiomyocytes. Authors state that “However, declines in 299 spontaneous contractile activity can occur if the conditions are maintained for more than 300 3 consecutive days, perhaps because PC-CMs are less mature than primary adult cardio- 301 myocytes, so the cells must be periodically returned to glucose-containing medium.” Since it is normally anticipated that more mature cardiomyocytes would lose spontaneous contractile activity, the authors’ above statement is misleading. The authors should clarify on this discrepancy with appropriate scientific evidence (specifically, why the cells should be returned to glucose-containing medium?)

Response: We have modified the indicated section of the manuscript (lines 330-332 “ However, prolonged exposure (e.g., >3 consecutive days) to low-glucose conditions can lead to substantial declines in cell viability and contractile activity, so the cells must be periodically returned to glucose-containing medium.”

  1. The authors mention in the abstract that this review manuscript will focus primarily on iPSCs in the context of cell therapy for heart failure. But, the authors have not included the currently ongoing clinical trials with human iPSC-cardiomyocytes, while they cited many clinical trials with other stem cell types. If the authors could cover current status of the iPSC-CM based clinical trials, that will greatly strengthen this review manuscript.

Response: No clinical trials of iPSC-CMs have been completed, but we have added a brief description of a clinical trial that has been initiated by researchers at Osaka University to our discussion of Recent Clinical Trials (lines 607-614).

Round 2

Reviewer 2 Report

The authors improved the manuscript, but some inaccuracies still left.

  1. The second sentence of the abstract (line 10) should be correct. The authors should carefully check English language.  
  2. Figure 2 is not mentioned in the text. The legend of the Figure 2 does not include extensions of abbreviations.
  3. It is too strong to state that „ASC are fundamentally different from the MSCs….” just because of CD146 expression. The CD146 is not obtained (or very little) in adult human skeletal myoblasts, human heart tissue-derived MSC or other types of adult tissue-derived MSC. It depends on how (or using what method) the ASC are isolated.
  4. The chapter (277-294 should be checked). Such expressions as: “…regulator of cardiomyocyte differentiation [79], “…analyses performed during four stages of murine and human ESC-CM differentiation“ or „…govern PC-CM differentiation [80]“ should be corrected, since cardiomyocytes do not differentiate to other types of the cells.
  5. Section 4.2 – it is unclear what type and origin of the cells the authors are talking about.
  6. In the same chapter – CD90 and CD140b are not suitable for all three types of the cells (e.g., “CD90 and CD140b for fibroblasts, smooth-muscle cells, and endothelial cells [58]”)?
  7. Section 4.3 is also not clear what type of the cells the authors are talking about.
  8. In vitro and in vivo or in situ should be written in Italic, not like in-vitro, in-vivo or in-situ. Check this out throughout the article.
  9. The names of genes should be in Italic, check this out throughout the article (for example, chapter 6 “as well as increases in the expression of ion channel (Kir2.1, HCN1, SCN5A, Kv4.3), calcium handling (CSQ2, junctin, triadin, SERCA), structural (Cav3, Amp2), and contractile (myosin heavy chain, myosin light chain) proteins [92].“ or „markers for cardiomyocyte maturation (TNNT2 and MYH6/7).“                        410 line "patch administration also increased the expression of markers for cardiomyocyte maturation (TNNT2 and MYH6/7). Are the authors talking about genes? If yes, the names of genes should be in Italic. The term  “Expression” is used for genes, not for proteins. Check this out throughout the article.
  10. 422 line –should be “electromechanically integrated within the patient’s intact…“, 424 line „that were administered as combination“.
  11. 481 line – the end of the sentence “provided the supply hiPSC-derived cells becomes suitably large and immunologically diverse.” is not clear.
  12. Table 2 does not have extensions of abbreviations. It should be explained below the Table.
  13. GTMMN needs an extension (494 line).
  14. Table 3 does not have extensions of abbreviations.

Author Response

Dear Reviewer,

We appreciate your very careful review and constructive input into our paper. We have addressed all the critique and we hope you will be satisfied with our revised version

Please find below a point-by -point response to the review

  1. The second sentence of the abstract (line 10) should be correct. The authors should carefully check English language.  

Response: Thank you very much for the carefully review and the typo has been Corrected.

  1. Figure 2 is not mentioned in the text. The legend of the Figure 2 does not include extensions of abbreviations.

Response: Figure 2 is mentioned on line 258, and the legends have been added.

  1. It is too strong to state that „ASC are fundamentally different from the MSCs….” just because of CD146 expression. The CD146 is not obtained (or very little) in adult human skeletal myoblasts, human heart tissue-derived MSC or other types of adult tissue-derived MSC. It depends on how (or using what method) the ASC are isolated.

Response: Thank you for pointing out. The intent to mention CD146 was to give an example of a marker that differs ASC from MSCs of other sources, among many other markers. “fundamentally” was removed.

  1. The chapter (277-294 should be checked). Such expressions as: “…regulator of cardiomyocyte differentiation [79], “…analyses performed during four stages of murine and human ESC-CM differentiation “ or „…govern PC-CM differentiation [80]“ should be corrected, since cardiomyocytes do not differentiate to other types of the cells.

Response: Cardiomyocyte differentiation and the rest simply means differentiation from stem cells to cardiomyocyte. It is a term that has been used in many other publications, notably references 79 and 80.

  1. Section 4.2 – it is unclear what type and origin of the cells the authors are talking about.

Response: The title of section 4 is named purification of iPSC-CMs. Adjustments were made to make it clearer.

  1. In the same chapter – CD90 and CD140b are not suitable for all three types of the cells (e.g., “CD90 and CD140b for fibroblasts, smooth-muscle cells, and endothelial cells [58]”)?

Response: According to the reference 58, negative selection targeting CD90 and CD140b can remove contaminating cells, such as fibroblasts, smooth-muscle cells and endothelial cells, from iPSC-CM culture.

  1. Section 4.3 is also not clear what type of the cells the authors are talking about.

Response: The title of section 4 is named purification of iPSC-CMs. Adjustments were made to make it clearer

  1. In vitro and in vivo or in situ should be written in Italic, not like in-vitro, in-vivo or in-situ. Check this out throughout the article.

Response: This issue has been addressed accordingly. Thank you.

  1. The names of genes should be in Italic, check this out throughout the article (for example, chapter 6 “as well as increases in the expression of ion channel (Kir2.1, HCN1, SCN5A, Kv4.3), calcium handling (CSQ2, junctin, triadin, SERCA), structural (Cav3, Amp2), and contractile (myosin heavy chain, myosin light chain) proteins [92].“ or „markers for cardiomyocyte maturation (TNNT2 and MYH6/7).“     410 line "patch administration also increased the expression of markers for cardiomyocyte maturation (TNNT2 and MYH6/7). Are the authors talking about genes? If yes, the names of genes should be in Italic. The term “Expression” is used for genes, not for proteins. Check this out throughout the article.

Response: Adjusted accordingly.

  1. 422 line –should be “electromechanically integrated within the patient’s intact…“, 424 line „that were administered as combination“.

Response: Adjusted.

  1. 481 line – the end of the sentence “provided the supply hiPSC-derived cells becomes suitably large and immunologically diverse.” is not clear.

Response: We intended to indicate that HLA typing could be used only if there are enough supply from different donors, just like in organ transplantation.

  1. Table 2 does not have extensions of abbreviations. It should be explained below the Table.

Response: Done

  1. GTMMN needs an extension (494 line).

Response: As noted in the lines 493-494 “GATA4, TBX5, MEF2C, MYOCD, NKX2-5; i.e., the GTMMN protocol”, it is an abbreviation for the 5 factors mentioned above.

  1. Table 3 does not have extensions of abbreviations.

Response: Done